

# Predicting RNA secondary structure by a neural network: what features may be learned?

Elizaveta I. Grigorashvili[1,*], Zoe S. Chervontseva[2,*] and Mikhail S. Gelfand[1,2]

[1] Center of Molecular and Cellular Biology, Skolkovo Institute of Science and Technology, Moscow, Russia
[2] Institute of Information Transmission Problems, Moscow, Russia
* These authors contributed equally to this work.

## ABSTRACT

Deep learning is a class of machine learning techniques capable of creating internal representation of data without explicit preprogramming. Hence, in addition to practical applications, it is of interest to analyze what features of biological data may be learned by such models. Here, we describe PredPair, a deep learning neural network trained to predict base pairs in RNA structure from sequence alone, without any incorporated prior knowledge, such as the stacking energies or possible spatial structures. PredPair learned the Watson-Crick and wobble base-pairing rules and created an internal representation of the stacking energies and helices. Application to independent experimental (DMS-Seq) data on nucleotide accessibility in mRNA showed that the nucleotides predicted as paired indeed tend to be involved in the RNA structure. The performance of the constructed model was comparable with the state-of-the-art method based on the thermodynamic approach, but with a higher false positives rate. On the other hand, it successfully predicted pseudoknots. t-SNE clusters of embeddings of RNA sequences created by PredPair tend to contain embeddings from particular Rfam families, supporting the predictions of PredPair being in line with biological classification.

## INTRODUCTION

Deep learning holds an important place in the toolbox of bioinformatics methods, applied to, *e.g.*, analysis of medical images (*Giger, 2018*; *McCallum et al., 2019*), classification of cancer subtypes (*Courtiol et al., 2019*; *Minnoye et al., 2019*), annotation of bacterial genes (*Clauwaert, Menschaert & Waegeman, 2019*), prediction of antibiotics (*Stokes et al., 2020*), prediction of ribosome stalling sites (*Zhang et al., 2017*), *etc*. Moreover, deep learning models can serve as a source of biologically meaningful insights. Basset, a convolutional neural network that predicts cell type-specific open/closed chromatin state given DNA sequence learned known and novel transcription factor-binding motifs (*Kelley, Snoek & Rinn, 2016*). DeepBind, a tool for determining sequence specificities of DNA- and RNA-binding proteins, captures local interactions in biological sequences (*Alipanahi et al.,*

Corresponding author
Elizaveta I. Grigorashvili,
elizaveta.grigorashvili@skoltech.ru

*2015*). These and other examples show that deep learning models can find short-distance interactions and reveal new, unknown motifs. However, the question of whether these models can capture non-local interactions was open until the first articles presenting neural networks predicting RNA structure appeared.

RNA secondary structure without pseudoknots can be predicted with a reasonable accuracy by minimizing the free energy ΔG using dynamic programming, as suggested by Zuker (*Zuker & Stiegler, 1981*). The crucial, experimentally determined parameters of the Zuker algorithm are the stacking energies of adjacent base pairs (that implicitly account for the energy of formation of the Watson–Crick and wobble pairs) and the destabilizing contribution of various types of loops. The prediction quality of the Zuker and similar algorithms is about 60–80%, measured by sensitivity and positive predictive value (PPV) (*Singh et al., 2019*). Sensitivity is a fraction of base pairs in the known structure that were predicted, and PPV is a fraction of correctly predicted base pairs. However, the performance of the algorithm deteriorates as the RNA length increases—predictions for sequences shorter than 800 nucleotides have the average sensitivity of 74% and PPV of 66% (*Mathews, 2004*) while, for instance, the average sensitivities for full-length small and large subunit rRNAs are 47% and 56% respectively (*Bellaousov & Mathews, 2010*).

Another concern about energy-minimization approaches is that the globally minimal-energy structure is not necessarily the correct one, as the actual state may correspond to a suboptimal structure with a comparable free energy. Such structure may be stabilized by interaction with small molecules, as in riboswitches (*Vitreschak et al., 2004*), tertiary interactions (*Martick & Scott, 2006*), and form co-transcriptionally (*Lai, Proctor & Meyer, 2013*). In addition, the free energy minimization is an NP-complete problem if pseudoknots are allowed (*Lyngsø & Pedersen, 2000*), and is highly sensitive to small changes in the energy parameters, in particular, those of loops (*Crowther et al., 2017*). Homology-based approaches proved to be useful (*Zhao et al., 2021*) but may not be applied to single sequences.

Deep learning-based techniques aim to overcome the described issues. Two major types of neural networks applied in biology are convolutional neural networks (CNNs), initially developed for image recognition, and recurrent neural networks (RNNs), typically used for language processing and speech recognition. The Attention mechanism is also popular for tasks where capturing context-dependent features is desired.

CNNs use a convolution operator to process the data coming in the form of arrays (*LeCun et al., 1989*). A convolutional layer consists of a set of learnable filters. Each filter convolves across the width and height of the input array, producing a two-dimensional activation map. Being stacked along the depth dimension, the maps from different filters give the output with the generalized features of the processed input. RNNs have internal states which retain information coming as a sequence of inputs. In particular, long short-term memory (LSTM) allow the network to learn long-term dependencies. A LSTM unit contains a memory cell and input, output, and forget gates regulating whether the received information should be retained and passed further, or should be reset (*Hochreiter & Schmidhuber, 1997*). The Attention mechanism may be applied to search for positions in the input with the most relevant information; this information is added as a context vector

to the output at each time step, so that the layers processing the Attention output are informed about the importance of input features (*Vaswani et al., 2017*). Weights defining the importance can be visualized and provide insights into the features used by the network for the prediction.

The first neural networks predicting the RNA structure, SPOT-RNA and DMfold, were both RNNs but differed in details of architecture and training. In SPOT-RNA, a RNN was enforced with pre-training on RNA spatial structures (*Singh et al., 2019*). DMfold does not use structural data but combines deep learning and dynamic programming, the latter implementing the base pair maximization principle (*Wang et al., 2019*). A recently published algorithm MXfold2 (*Sato, Akiyama & Sakakibara, 2021*) combines a neural network and the thermodynamic regularization. Folding scores predicted by MXfold2 are strongly correlated with actual free energy; however, without the thermodynamic regularization the correlation is lost.

Notably, none of these papers discuss the features used by the neural networks for the prediction. The analysis of features created by deep learning models is known as the *representation learning* (*Goodfellow, Bengio & Courville, 2016*). The aim is to reveal the internal logic of the neural networks' decisions and thus design better models. In an attempt to understand what is learnable in this setting, we created a neural network trained on RNA sequences only, ignorant about any additional information like thermodynamic parameters, spatial organization, or base pair maximization principle, and analyzed the biological meaning of the network predictions and its internal representations. While this network based on a data-driven approach naturally performs worse than specialized models relying on prior information about the domain, the learned features proved to be biologically meaningful, demonstrating that this approach may in general contribute to understanding of the underlying phenomena.

## MATERIALS AND METHODS

### Model architecture

Two parallel convolutional layers first processed the input with 64 filters and the kernel size of 10. The convolved input and the original input were then transformed by the Attention layer (*Vaswani et al., 2017*). The result of the Attention and the original input was concatenated and processed by the bidirectional LSTM with 16 neurons (*Hochreiter & Schmidhuber, 1997*). The LSTM output is conveyed to two dense layers of 16 and eight neurons and then to a dense neuron wrapped into the TimeDistributed layer. The latter had L2 regularization for both kernel and bias terms with the learning rate of 0.1. The Softmax activation was used in the output neuron to normalize its input value to the range [0, 1] and to sum the output values to 1. To adjust learning, we used the Adam optimizer (*Kingma & Ba, 2017*). The loss function was categorical cross-entropy, and the performance was estimated using an accuracy metric.

Additionally, we made preliminary tests on artificially generated random sequences containing 7-nucleotide-long complementary stretches. PredPair, weights, and data are available *via* the link: https://github.com/octolis/PredPair.

## Dataset and training

We used RNA sequences from seed alignments of 2,147 Rfam families (*Kalvari et al., 2018*). This set did not include families having less than 20% of paired nucleotides. We used one-hot encoding to process the data for the neural network. For each sequence, we generated a set of "question samples" and "answer samples": in the question sample one and only one position was marked, and the only pairing for the question was marked in the answer sample. To include this structural information in the encoding, we added the fifth dimension to the vector containing '2' for the query position in the "question sample" or the correct pairing in the "answer sample" and '1' for all remaining positions. This procedure led to 857,307 pairs of "questions" and "answers" that were organized in batches each representing one particular sequence. The dataset was split into the train, validation, and test sets in the ratio of 0.6, 0.2, and 0.2, respectively. The split on train, validation, and test was performed in so that all members of one Rfam family were assigned to one set.

## Symmetrization of the base-pairing certainty matrix

By design, the answer of the network sums to 1 for each row, which represents the probabilities of nucleotides to be paired with the query nucleotide. It means that the base-pairing certainty matrix $M$ is mostly asymmetric and $M(i,j) \neq M(j,i)$. This asymmetry contradicts the notion of "being paired" (if $i$ is paired to $j$, then $j$ must be paired to $i$). $M(i,j)$ is the probability that nucleotide $i$ will select nucleotide $j$ out of possible variants. $M(j,i)$ is the probability that nucleotide $j$, in its turn, will select nucleotide $i$ from all its variants. As these events are not independent, it would be incorrect to multiply the probabilities, and we take the minimum instead. Hence, we defined the score matrix $S$ such that $S(i,j) = S(j,i) = min\ (M(i,j),\ M(j,i))$ to force $M$ to be symmetric. As we only need to compare the values in one row, there is no need to introduce a scaling coefficient to force the sum of elements in each row to be 1.

Moreover, as shown further in the section of the Results devoted to the description of the model and data used, the true partners are more likely to be the best bidirectional hits for each other. So, the absolute value of the minimum provides an additional information on whether the nucleotide is likely to be paired.

## Feature importance analysis

To assess the importance of each nucleotide in the prediction of base-pairings, we used a gradient-based measure (Vanilla Gradient) of how the change in the input values affect the answer (*Simonyan, Vedaldi & Zisserman, 2014*). The Vanilla Gradient is an algorithm for the construction of saliency maps. It calculates the gradient of the loss function for the output class with respect to the input data, yielding a heatmap of the size of the input features. The values of the heatmap reflect the importance of each input feature for the prediction. Here, gradient-based saliency maps were computed using TensorFlow GradientTape (a class for the recording of operations for automatic differentiation). Each resulting feature importance matrix was symmetrized as described above for the base-pairing certainty matrix.

## Comparison with DMS-seq data

We downloaded the genome of the reference strain *E. coli* MG1655 (NC000913.2) and selected all coding sequences on the plus chain. To eliminate pseudogenes, we excluded sequences annotated other than "CDS", and those with the length not divisible by three. We predicted the pairings using PredPair and selected pairs of nucleotides which were the best bidirectional hits for each other. To obtain the experimental data to compare the predictions with, we downloaded data on the accessibility of nucleotides (DMS-Seq) for this strain from GEO (accession GSM2055260; *Burkhardt et al., 2017*). All coding sequences that had coverage more than 15 reads per nucleotide in the DMS-Seq data were used. The values of accessibility in the DMS-Seq data were split to deciles separately for each gene, sorted from the least accessible to the most accessible. The predictions by PredPair were mapped to the deciles by calculating how many nucleotides predicted as bidirectionally paired belonged to each decile in the DMS-seq data; the results were plotted as a histogram. The same procedure was performed with RNAplfold and SPOT-RNA tools for comparison.

## Comparison with RNAplfold and SPOT-RNA

RNAplfold is a tool for prediction of base pair probabilities averaged within a window size (*Lorenz et al., 2011*). This program uses the Zuker algorithm for prediction. We took all the sequences from the test dataset and predicted their base-pairings using RNAplfold with the following parameters: cutoff (lower threshold value of the probability) 0.0, window length equal to the length of the longest sequence. Then we calculated a confusion matrix for this prediction with the Rfam secondary structure annotation taken as the ground truth and used the confusion matrix to plot the precision–recall curve. The procedure for constructing the curve was as follows. We set a certain threshold value for a certainty of the prediction. With a fixed threshold value, we calculated the matrix of True Positives (Tp), False Positives (Fp), and False Negatives (Fn), and used them to calculate the precision = $Tp/(Tp + Fp)$ and recall = $Tp/(Tp + Fn)$. The obtained values represented a single dot for the precision–recall curve. This procedure was repeated 200 times with different threshold values and provided the data to plot the curve. The same procedure with the Rfam secondary structure annotation as the ground truth was used to obtain the precision-recall curve for PredPair and SPOT-RNA predictions. The calculated values for precision and recall were also used to calculate F1 scores for individual sequences for all three methods.

## RNA sequence embeddings

RNA sequence embeddings were created by taking activations of a batch of "question" sequences (representing one particular RFAM sequence) after the biLSTM layer and averaging them along the sequence. This procedure generated a vector of length 32 for each sequence. For each family, 100 sequences were randomly sampled for clustering. t-SNE was performed by using the class sklearn.manifold.TSNE from the Python library scikit-learn (version 0.23) with default parameters (number of components of 2, perplexity of 30.0, learning_rate of 200.0, number of iterations of 1,000).The description of the

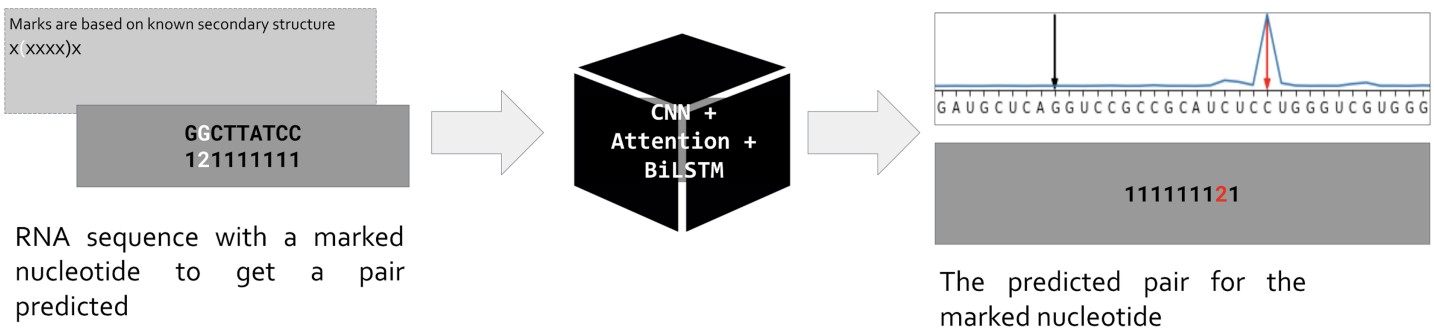

**Figure 1 Problem setup.** PredPair, a neural network (center) takes as input RNA sequence with one nucleotide marked by '2' (left, dark grey) and predicts a position where the pair of the marked nucleotide is located (right, dark grey). The marks are based on known RNA secondary structure (light grey box on the left). The main layers of PredPair are shown over the black box in the middle: CNN, convolutional neural network; Attention, Attention layer; BiLSTM, bidirectional long short-term memory layer.

parameters is available *via* the link https://scikit-learn.org/0.23/modules/generated/sklearn.manifold.TSNE.html.

## RESULTS

### Model and data

Our approach is based on the standard assumption that an RNA sequence is necessary and sufficient for the RNA folding. Hence if we train a neural network to predict the position of a pair for a given nucleotide, the network prediction will be based on features hidden in the RNA sequence (Fig. 1). As input, we use one-hot encoded RNA sequence with one marked nucleotide, and we expect the network to predict the position of a pair for the marked nucleotide. So, the input is a matrix $(L, 5)$ where $L$ is the sequence length, and the dimensionality five includes one-hot encoded RNA of the form $(L, 4)$ and an additional vector of length $L$ for the mark. The marks are based on the known secondary structure taken from Rfam seed alignments (*Kalvari et al., 2018*). Only nucleotides involved in pairings are marked: during the training, we do not ask the network to decide about unpaired positions. We put marks for each pair $(i, j)$ in both directions, so each pair generates two inputs, one with marked position $i$ (expecting to predict position $j$), and vice versa. Thus, the total number of input matrices equals to the number of positions involved in canonical and wobble pairing over the entire dataset.

The network outputs a vector of length $L$ that assigns a value from 0 to 1 to each position. We call this value *the certainty (of the prediction)*, and here and below *prediction* simply means the position of the output vector with the highest certainty (Fig. 1, upper right corner). Stacking all output vectors for a given RNA sequence, we obtain a matrix which we refer to as the *base-pairing certainty matrix*.

As RNA secondary structure is determined by the sequence, but involves distant, complex relationships, we tested several architectures, finally settling on an architecture containing CNN as a feature extractor processing input in a layered manner, capturing complex relationships (*Khan et al., 2020*), LSTM for sequential data processing that spreads the input information along the sequence (*Jurtz et al., 2017*), and Attention to

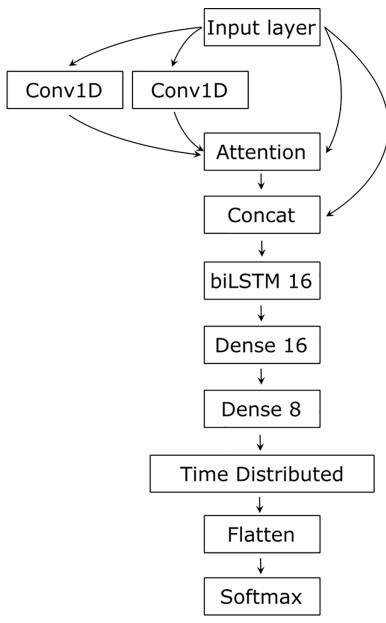

**Figure 2 The network's architecture.** Conv1D, one-dimensional convolution layer; Attention, Attention layer; Concat, Concatenation layer; biLSTM 16, 16 bidirectional LSTM neurons; Dense 16 and Dense 8, 16 and 8 dense neurons with ReLU activation function; TimeDistributed, TimeDistributed wrapper with ReLU activation; Flatten, flatten function; Softmax, Softmax layer.

integrate the input and establish and visualize the relationships between its parts (*Vaswani et al., 2017*) (Fig. 2).

To ensure this combination of neural network layers can catch distant interactions in sequential data, we made preliminary tests on less complex data, artificially generated random sequences containing seven-nucleotide-long complementary stretches. These tests showed that this architecture can capture long-distance complementary interactions and identify pairs in RNA, as prediction accuracy was close to 100% (data not shown).

To train, test, and evaluate PredPair, we used the largest possible dataset of RNAs from all known Rfam families (*Kalvari et al., 2018*). The dataset consisted of RNA sequences from seed alignments of all families in the Rfam database. The train/validation/test split was 60/20/20: sequences from 1,311 Rfam families in the train set, sequences from 412 families in the validation set, and sequences from 424 families in the test set. Splitting by Rfam families aimed to exclude the homology between training and test set sequences and hence, data leakage (*Jones, 2019*).

To evaluate the PredPair performance, we calculated top-1 and top-2 accuracies (that is, how often a correct answer corresponds, respectively, to one or two highest values of the output vector). The top-1 accuracy was 0.58, and the top-2 accuracy, 0.70.

Some erroneously predicted positions had rather high certainty values. However, these errors were not symmetrical: if a certain position $i$ had an erroneously predicted pair $j_{err}$, the highest-scoring pair for $j_{err}$ was often not $i$. This contrasts with the majority of correct predictions that were reciprocal: if position $j$ was predicted to pair with position $i$, the prediction for $j$ was likely to be $i$ (Fig. 3).

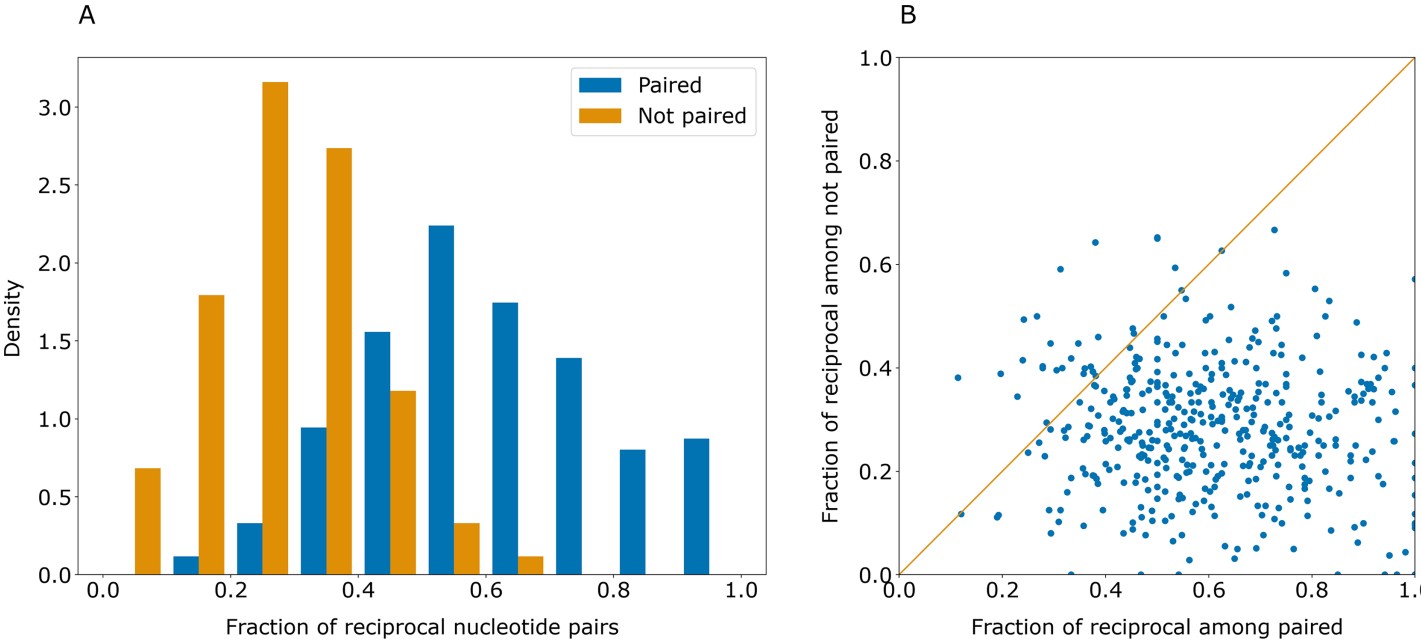

**Figure 3** **Paired positions tend to be predicted symmetrically.** (A) Distributions of the fraction of reciprocally predicted nucleotides for actually paired nucleotides (blue) and erroneously predicted pairs (orange). (B) Distribution of fractions of reciprocally predicted nucleotides out of actually paired (horizontal axis) and unpaired nucleotides (vertical axis) for individual sequences (dots).

As correct pairs are predicted reciprocally more often than incorrect ones, the closeness of the values in the cells of the base pair certainty matrix for each pair may be considered as the level of confidence in the prediction. After transformation of the base pair certainty matrix to contain the minimum of $(i, j)$ and $(j, i)$ certainty values for each pair of symmetric positions, the top-1 accuracy of PredPair on the test set increased to 0.63.

To benchmark PredPair, we compared the result with the widely used method RNAplfold (*Lorenz et al., 2011*) and deep learning based tool SPOT-RNA(*Singh et al., 2019*). In RNAplfold, Zuker's algorithm is implemented to predict the probabilities of base pairs in a given RNA molecule. We compared the Precision-Recall metric and distributions of F1 scores for individual sequences for PredPair, RNAplfold, and SPOT-RNA (Fig. 4). For this comparison, the pairs of known Rfam structures were considered as a positive set, whereas the remaining pairs were assigned to the negative set. PredPair did not outperform RNAplfold and SPOT-RNA mainly due to lower sensitivity and higher false positives rate. The Precision-Recall curve of SPOT-RNA suggests that this method at some range of thresholds is more precise than PredPair and RNAplfold, yet misses true positives (Fig. 4A). However, the distributions of F1 scores indicates that the trade-off between precision and recall is comparable for PredPair and SPOT-RNA (Fig. 4B). As our aim was to find biologically meaningful features learned by the network, we considered the comparison results to be sufficiently good to warrant further analysis of the predictions.

**The model's predictions are consistent with experimental data**
As no coding sequences were used for training PredPair, we used the data on nucleotide accessibility of mRNAs in *E. coli* to validate the results on independent experimental data

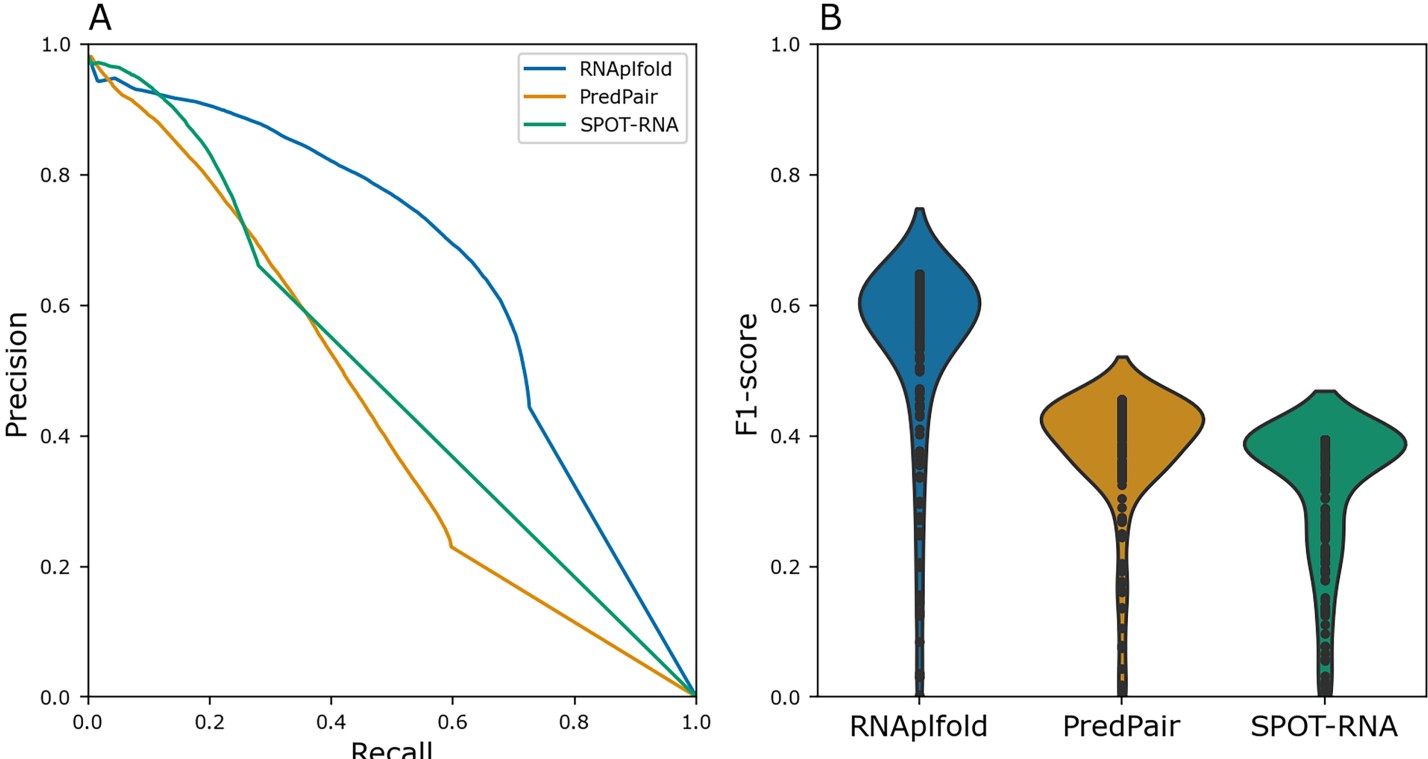

**Figure 4 Performance quality.** (A) Precision-recall curves for RNAplfold (blue), PredPair (orange), and SPOT-RNA (green) generated from the confusion matrices calculated with different thresholds for predictions to be accepted as true ones (see Methods for details). The curves were obtained on the test set data. (B) Distribution of F1-scores for RNAfold (blue), PredPair (orange), and SPOT-RNA (green) calculated from the precision and recall values.

obtained using the DMS-seq technique (*Burkhardt et al., 2017*). Dimethyl sulphate (DMS) reacts with unpaired adenines and cytosines and hence reveals the accessibility of these nucleotides. PredPair, in its turn, assigns values reflecting a certainty of being paired with the marked nucleotide, thus, predicts possibly non-accessible positions. We hypothesized that positions of nucleotides least accessible by DMS-seq should often be predicted as paired by PredPair.

We asked PredPair to predict pairings in mRNA sequences from the DMS-Seq experiment (GEO accession: GSM2055260, *Burkhardt et al., 2017*) and selected pairs of nucleotides which were the best bidirectional hits for each other. We sorted nucleotides in the experimental data by decrease of their measured accessibility and mapped positions of predicted paired nucleotides on the resulting deciles. The same procedure was performed with RNAplfold and SPOT-RNA. The distribution of predicted nucleotides over deciles of positions in the experimental data shows a clear trend: nucleotides predicted to be paired tend to be less accessible in the experiment (Fig. 5). PredPair (A), RNAplfold (B) and SPOT-RNA (C) exhibit the tendency to predict nucleotides involved in secondary structure in experiment as paired. The tendency for PredPair is more pronounced than for RNAplfold ($p$ value of $6.8 \times 10^{-58}$ for the Mann-Whitney test). The difference between PredPair and SPOT-RNA distributions is significant ($p = 3.5 \times 10^{-14}$ for the Mann-Whitney test); given high precision with relatively lower recall of SPOT-RNA, it is

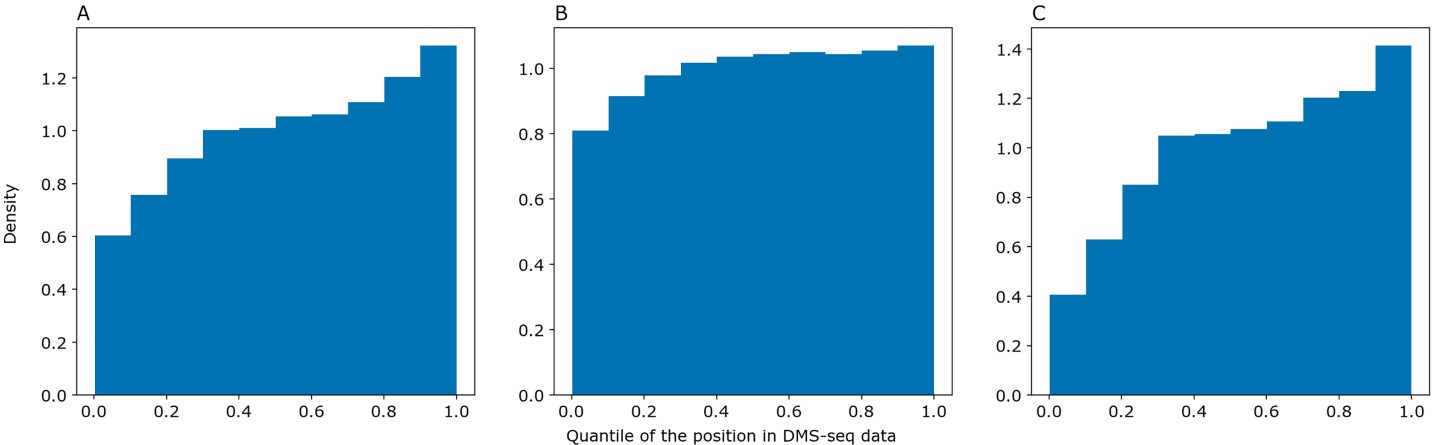

**Figure 5 Comparison of predicted and experimental structure in mRNAs.** Comparison of predicted and experimental structure in mRNAs. The distribution of nucleotides predicted as paired by PredPair (best bidirectional hit), (A), RNAplfold (B), and SPOT-RNA (C) over the deciles of positions in the DMS-Seq data.                                                                               

not surprising, as fewer false positives together with more false negatives shift the distribution.

### Some examples

We visualized predictions for several RNAs from the test set and compared them with the actual structure from the Rfam database and the prediction of RNAfold (*Lorenz et al., 2011*). As PredPair identifies base-pairing partners, we focus on the diagonal patterns of the matrices corresponding to known elements of the structure. Figure 6 shows the comparison for the DUF2693-FD RNA motif (Rfam accession: RF02926). This motif has a central loop surrounded by two hairpins and a stem (Fig. 6A). PredPair has successfully captured this structure: the certainty values for correct pairs are higher than for erroneously predicted ones (Fig. 6C). Here, RNAfold assigned higher probabilities to other basepairings, although captured the correct ones as suboptimal (Fig. 6B).

Figure 7 shows a similar comparison for a viral upstream pseudoknot domain (Rfam accession: RF01105). PredPair produced high certainty predictions not only of base pairs in the regular hairpin but also for pairs interacting in the pseudoknot. This finding is somewhat unexpected, as pseudoknotted structures comprise only a small fraction of the training dataset. We have not applied RNAfold here as it cannot predict pseudoknots. Interestingly, on pseudoknots (not present in the training set), the top-1 accuracy of PredPair was 0.78, that is, even higher than on average.

### Features learned by PredPair

To assess the importance of particular nucleotides for PredPair predictions, we performed the gradient-based analysis as described in Methods. As a result, for each structure we obtained a matrix of importance values for each pair of positions and compared it with the certainty matrix (Fig. 8). The most important positions are concentrated close to (1) the query nucleotide and (2) its partner.

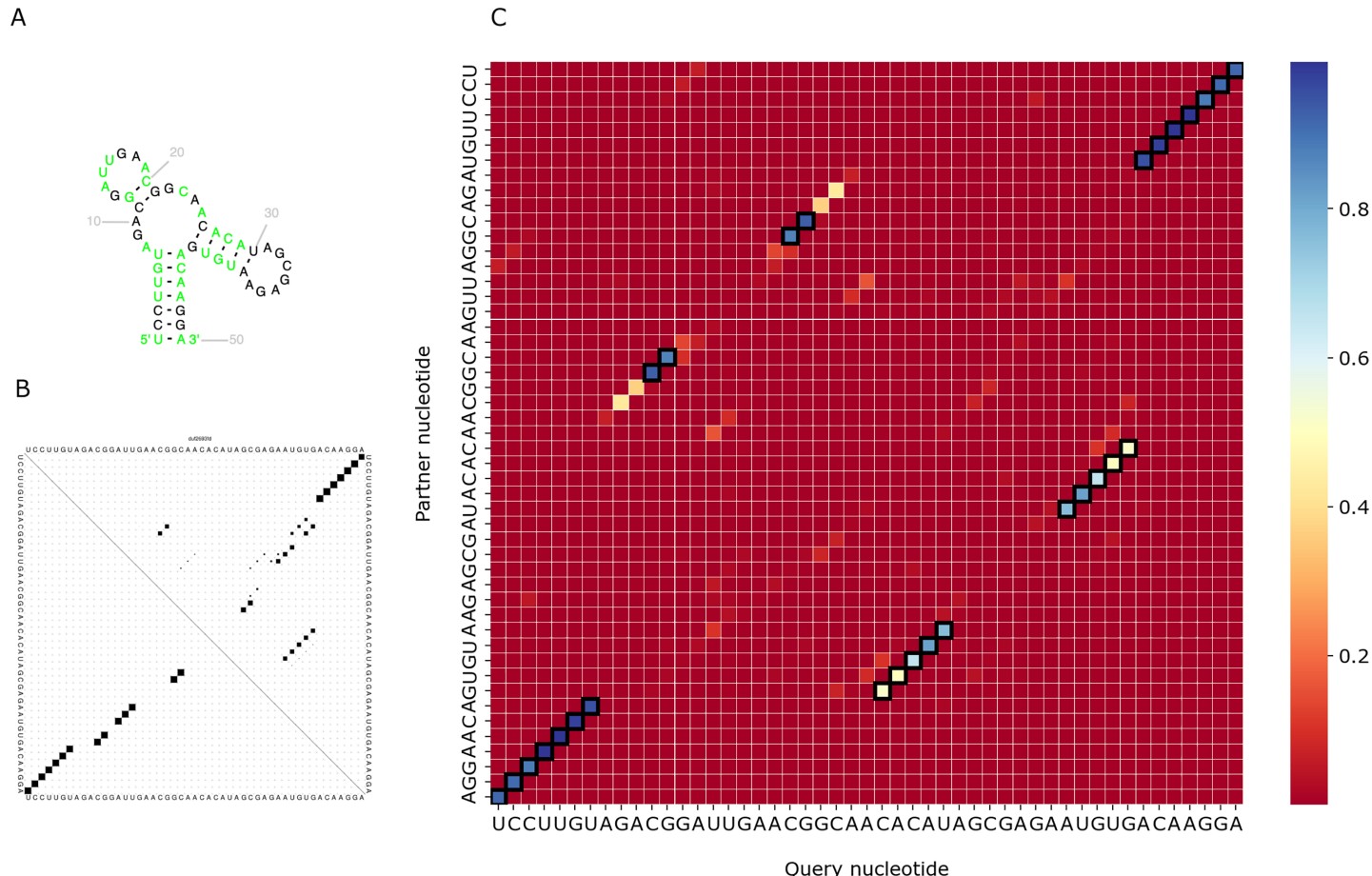

**Figure 6 Comparison of the DUF2693-FD RNA motif structures.** (A) The structure from the RNAcentral database (*RNAcentral Consortium,* *2021*). (B) The base-pairing probability matrix predicted using RNAfold. (C) The heatmap representing all base pairs predicted by PredPair. Black rectangles mark base pairs from the Rfam structure. Color represents the certainty of the prediction with the scale given aside. The matrix has not been symmetrized, so for each row, the sum of all values of the row is 1 (see Methods for details)

The importance measure tends to be larger for paired positions than for non-paired ones. This is consistent with the previously described symmetry: correct predictions tend to be reciprocal while incorrect predictions do not, and the same tendency is true for the importance measure. The confidence of the net in its answer, *i.e.*, the symmetrized base-pairing certainty matrix values (Fig. 9A) and the importance values (Fig. 9B) differ from the respective values for randomly selected nucleotides.

To determine whether PredPair managed to learn any energy-related features, we calculated the prediction frequencies of various pairs and adjacent pairs of stacked pairs. Indeed, PredPair predicts G-C pairs more often than A-U pairs, and A-U pairs more often than wobble G-U pairs (Fig. 10A). All remaining pairs are extremely rare. Moreover, PredPair tends to predict adjacent, stacking base pairs with less free energy more often than the ones with higher free energy: frequencies of stacked base pairs are negatively correlated with known stacking energies in Kcal/mol with Spearman $r$ of $-0.67$ and $p$-value of $8.9 \times 10^{-6}$ (Fig. 10B).

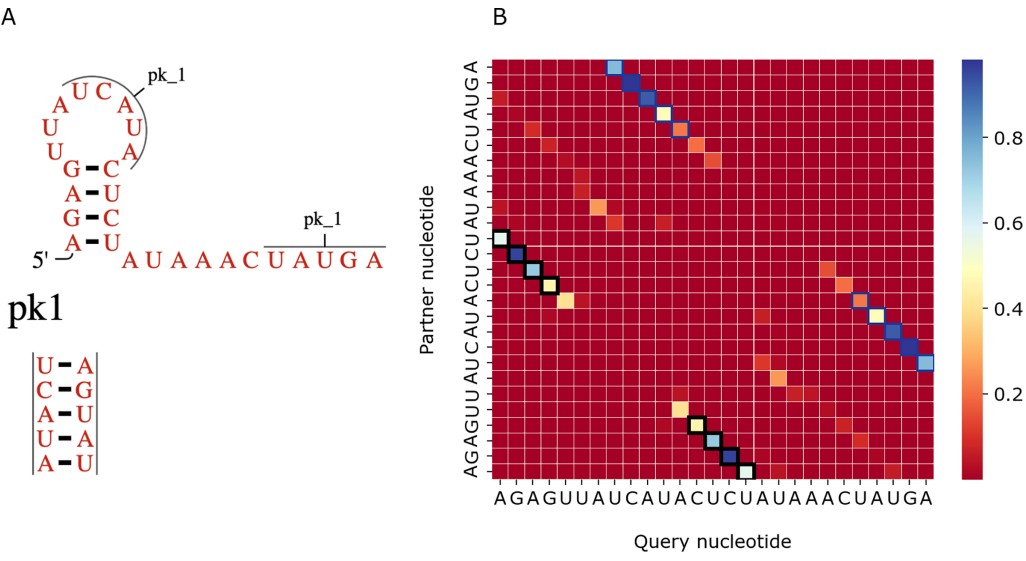

**Figure 7 Viral upstream pseudoknot domain structure.** (A) Rfam accession: RF01105. (B) The heatmap representing all base pairs predicted by PredPair. Notation as in Fig. 6; positions forming the pseudoknot are framed in blue.

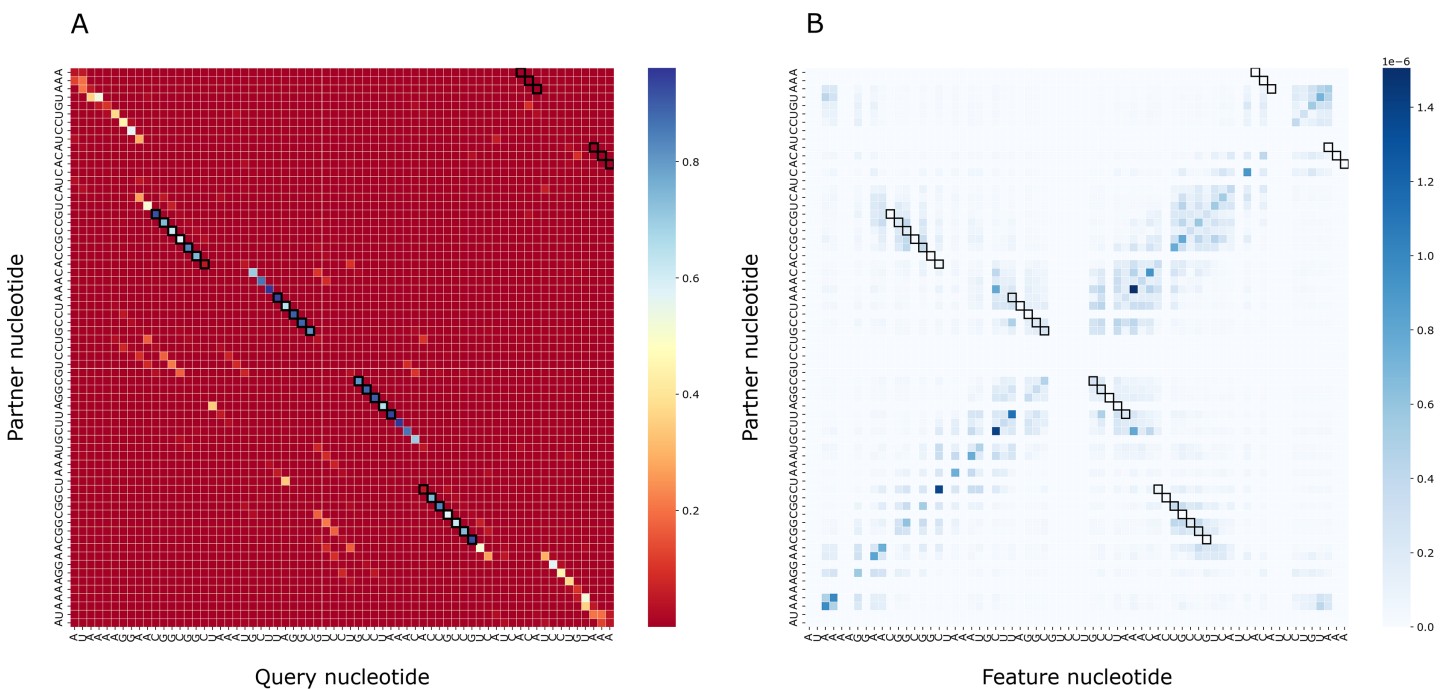

**Figure 8 Matrices of importance values for *Bacillus subtilis* subsp. subtilis str. 168, Bacillaceae-1 RNA (Rfam accession: RF01690).** (A) The heatmap representing all base pairs predicted by PredPair. (B) The heatmap representing the importance of positions for the prediction (see the text for details).

PredPair contains the Attention layer that should catch context-dependent features (*Vaswani et al., 2017*). Of five neurons in the Attention layer, four have higher weights at positions with specific nucleotides (A, T, G, C), and the fifth one tends to have higher

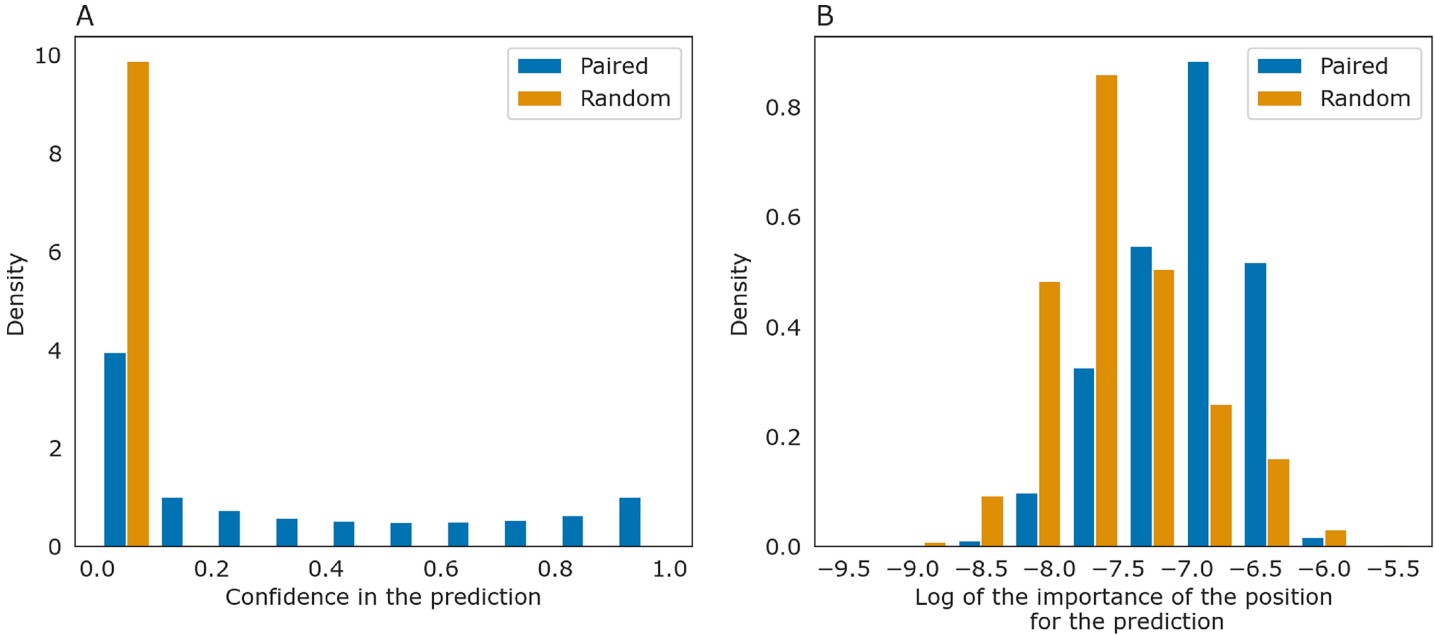

**Figure 9 Prediction confidence and feature importance for paired (blue) and random (orange) nucleotides.** (A) The distributions of the symmetrized certainty matrix value. (B) The distribution of logarithm of position importance for pairs having nonzero importance. The number of zeros is higher for random pairs (data not shown). In both cases, the distributions differ significantly (Mann-Whitney $p$-value $< 10^{-40}$).

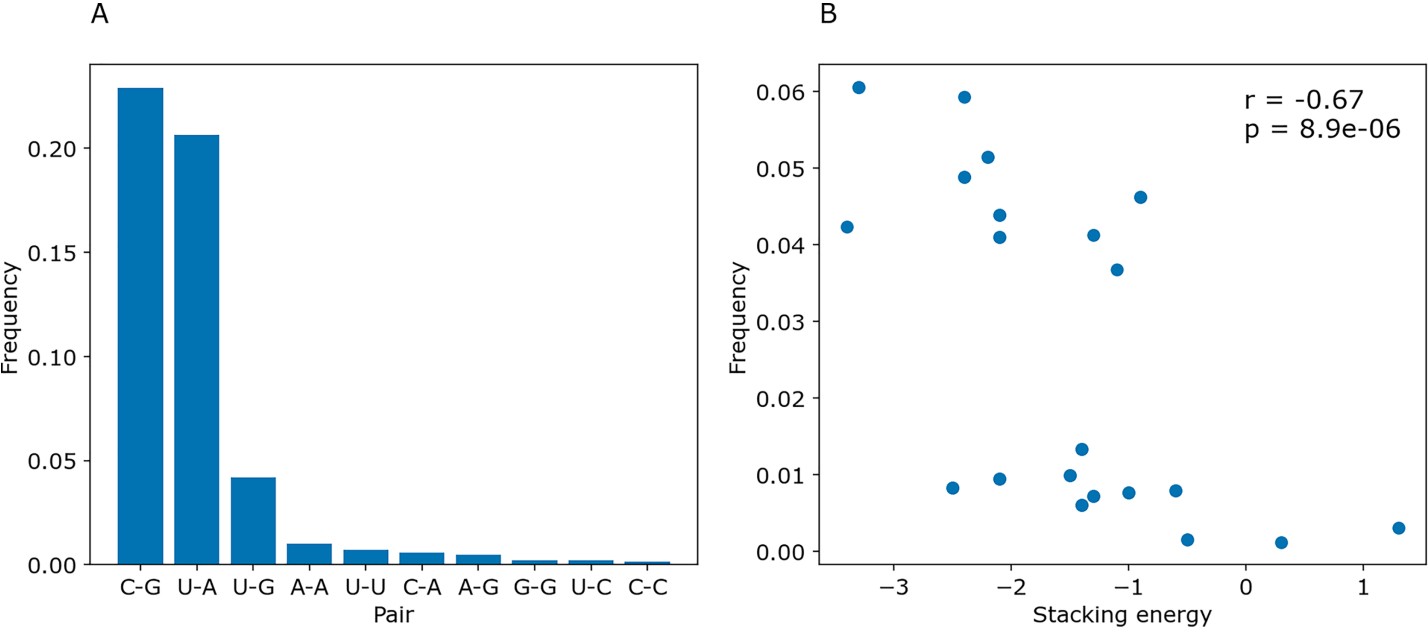

**Figure 10 Frequencies of predicted pairs and stacked pairs of base pairs. Predictions were made for *E.coli* coding gene sequences.** (A) Relative frequencies of base pairs. (B) Frequencies of stacked base pairs (vertical axis) and the stacking energies taken from (*Gorodkin & Hofacker, 2011*) (horizontal axis); each dot is a pair of stacked base pairs.

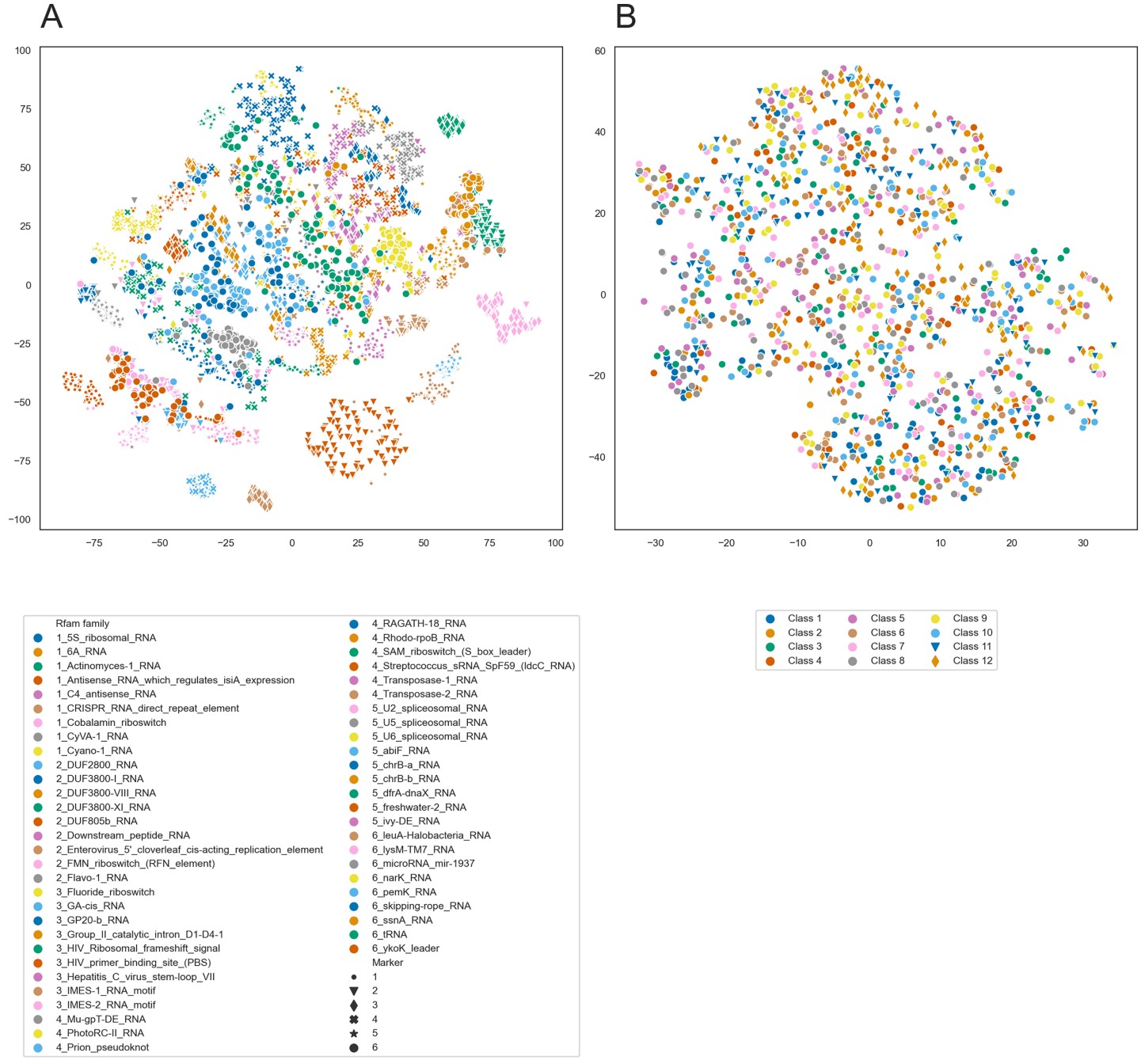

**Figure 11  t-SNE clustering of RNA embeddings.** (A) t-SNE plot for embeddings of 54 Rfam families created with PredPair. Due to restrictions in coloring, 54 Rfam families were randomly split into six groups assigned with four numbered shapes of the marker (the numbers are at the beginning of the Rfam families title in the legend). (B) t-SNE plot for embeddings of twelve sets of artificially generated sequences that may fold into 12 types of secondary structures (see the text for details of the generation of random sequences). Due to restrictions in coloring, dots representing sets 11 and 12 differ in shape.

weights for nucleotides complementary to the query one (data not shown). This corresponds to the design of one-hot encoded input vectors of the form (L, 5). Vectors 1–4 contained information about the encoded letter, and the corresponding neurons 1–4 have

higher weights for the respective nucleotides. Vector 5 contained the mark for the prediction of the pair, and indeed neuron five in the Attention layer accumulated weights at the positions of complementary nucleotides. These observations suggest that the Attention layer might not be a source of more complex features; at least, not in the present form.

To further analyze the internal representations learned by PredPair, we used it to create embeddings of RNA sequences. We selected 100 sequences from each of 54 Rfam families and created the embeddings from the sequences by averaging the activation weights from the biLSTM layer. Then, we clustered these embeddings using t-SNE and found that the sequences tend to cluster by the Rfam family, as shown in Fig. 11A.

The clustering could be explained in two ways. It may reflect either sequence similarity within Rfam families or structural features derived from the RNA secondary structure. To distinguish between these possibilities, we designed the following experiment. We generated random sequences such that they could adopt a given secondary structure and did it for several existing secondary structures. If the network has indeed learned the complex structural features, we could expect the embeddings created from the random sequences to form t-SNE clusters as did the embeddings of the real sequences. If not, the embeddings are likely not to be the source of information about RNA secondary structure.

We selected twelve random secondary structures of length 140 from Rfam. For each structure, we generated 100 RNA sequences that can potentially fold into this structure: for positions paired in the structure we generated nucleotides that can base pair. Selection of nucleotides was random but weighted on the frequencies of nucleotides, nucleotide pairs, and pairs of dinucleotides. The frequencies were calculated from the Rfam data. These 1,200 sequences were transformed into the embeddings using PredPair and clustered with t-SNE. t-SNE did not show any clustering for the artificial sequences (Fig. 11B), so the embeddings are likely not to be a source of information about the RNA secondary structure.

## DISCUSSION AND CONCLUSIONS

We aimed to access the ability of neural networks to find non-local interactions by sequence alone. Hence, we designed a neural network and training setup so as to minimize the input information and to avoid external, fixed parameters. The network knew only the sequence and, for training, the base pair partner for the query nucleotide. We did not provide data on other elements of the secondary structure, let alone 3D structures, nor did we incorporate information about allowed base-pairing or stacking energies. Still, PredPair learned the Watson-Crick and wobble base-pairing rules, developed an internal representation of the stacking energies, and showed the ability to adequately predict pseudoknots.

While the performance of PredPair is comparable with that of RNAplfold, it is understandably worse due to lower sensitivity and higher false positives rate, natural for an algorithm that has absolutely no *ad hoc* information. As PredPair is not an end-to-end model for the RNA structure prediction, we cannot compare it with published deep learning-based methods. However, the comparison of the predictions with the

experimental data provides an argument for the adequacy of PredPair predictions. Indeed, this comparison to the nucleotides' accessibility shows a clear tendency for the nucleotides predicted as paired to be less accessible in the natural RNAs. Notably, these results were obtained with mRNAs, the type of sequences that were not used in training and were completely new for PredPair.

The PredPair architecture allowed us to create embeddings—abstract representations—of RNA sequences. The t-SNE clusters of embeddings of sequences from the test set corresponded to Rfam families of these RNAs. However, as shown by the analysis of embeddings of random, structured sequences, there is a possibility that PredPair caught sequence, rather than structural patterns of Rfam families. Still, as there were no data leakage due to homology, and PredPair performed well when applied to independent DMS-seq data, PredPair did not simply learnt a set of sequence patterns. At that, while the information about the sequence patterns, complementarity, and tendency of complementary nucleotides to assemble into long stretches could be sufficient in a simple setup of finding a pair for a given nucleotide, more complex structural patterns could be learned by a network trained to solve more sophisticated tasks.

### Funding
This study was supported by the Russian Science Foundation under grant 18-14-00358 to M.S.G. The funders had no role in study design, data collection and analysis, decision to publish, or preparation of the manuscript.

### Grant Disclosures
The following grant information was disclosed by the authors:
This study was supported by the Russian Science Foundation under grant 18-14-00358 to M.S.G. The funders had no role in study design, data collection and analysis, decision to publish, or preparation of the manuscript.

### Competing Interests
Mikhail Gelfand is an Academic Editor for PeerJ.

### Author Contributions
- Elizaveta I. Grigorashvili conceived and designed the experiments, performed the experiments, analyzed the data, prepared figures and/or tables, authored or reviewed drafts of the article, and approved the final draft.
- Zoe S. Chervontseva conceived and designed the experiments, performed the experiments, analyzed the data, prepared figures and/or tables, authored or reviewed drafts of the article, and approved the final draft.
- Mikhail S. Gelfand conceived and designed the experiments, analyzed the data, authored or reviewed drafts of the article, and approved the final draft.

## Data Availability

The data and code are available at GitHub: https://github.com/octolis/PredPair/tree/master; octolis. (2022). octolis/PredPair: first release (v1.0). Zenodo. https://doi.org/10.5281/zenodo.7214860.

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
