# Peer review of "Predicting RNA secondary structure by a neural network: what features may be learned?"

_PeerJ, doi:10.7717/peerj.14335_

## Round 0.1 · original submission · Major Revisions

Your manuscript has been reviewed by two experts in the field. As you can see from their comments below, both of them raise fundamental points. Particularly, Reviewer 2 gives serious concerns on the design of your experiments. if you admit the validity of these concerns, perhaps you had better give up submitting its revision. But I would like to give you an opportunity to respond to the reviewers if you think that their comments are not appropriate. Please read the comments carefully. Although I will receive your revised manuscript, maybe it is difficult to persuade the reviewers and so please understand that the possibility of accepting your revised manuscript is not so high.

Reviewer 1 ·

Basic reporting

The authors developed a method for predicting RNA secondary structures using deep learning. Although its accuracy was not improved over existing methods, by interpreting learned results with gradient-based methods and by analyzing the embeddings, the authors showed that the learned results are biologically valid.

l.143: min(M(i,j), M(j,i))
Why take the minimum? By taking the minimum, the sum will not be 1, but please discuss how to interpret this.

l.148: gradient-based measture (Vanilla Gradient)
This method is not obvious and should be described in detail in this paper.

Experimental design

l.152-: Comparison with DMS-Seq data
Described in this section are the data used and how they were compared is missing.

l.161-: Comparison with RNAplfold
Similarly, this section lacks how they were compared.

l.169-: RNA sequence embeddings
It should describe how the tSNE was implemented and what parameters were used.

l.270: Features learned by PredPair
Here shows an analysis that is not written in the Method section. The method of the analysis being performed here should be described in the Method section.

l.304-311:
It is not clear what you want to say in this paragraph. Please rewrite it more carefully. Furthermore, you did not explain how you generated "specificially generated sequences with particular secondary structures". This should be written in the Method section.

Validity of the findings

l.247: The distribution of predicted nucleotides over ...
Could the same experiment be done with RNAplfold? If it can be done, I think a comparison is needed.

Reviewer 2 ·

Basic reporting

Grigorashvili et al attempted to develop a method for predicting RNA secondary structure. The paper provides an adequate review of current progresses in RNA secondary structure. However, it seems a bit strange about the way they designed for prediction and the poor accuracy of the resulting method.

Experimental design

The method was designed to pre-define a nucleotide that is involved in base pairing and then to predict the partner of the marked nucleotide. It is a bit strange to make a predictor like this. First, one can't call this method as a secondary structure predictor as it requires an input of bases involved in pairs. Second, it will have a limited usefulness as we are certainly don't know if a base is involved in pairing or not. The method will be only useful to apply to experimental probe results, which provided pairing profiles. However, these experimental probe results are not that accurate.

Validity of the findings

Nevertheless, the method was built on a good training and test sets as it used Rfam to reduce the possibility of the same family in the training and test sets. However, the resulting method is far less accurate than a typical secondary structure predictor RNAplfold. This raises the serious question why bother to use this method if RNAplfold can do it better in the absence of the pairing profile information.

---

## Round 0.2 · Major Revisions

First of all, I would like to explain why it took so much time in reviewing your revised manuscript: there was a split of opinions (acceptance or rejection) between the two original reviewers. Thus, I invited two more reviewers. As you can see from their comments below, both of the two new reviewers recommend major revision but Reviewer 3 thinks that another round of revision may not be able to address her/his concerns. Then, you may choose either to withdraw the submission or to re-revise the manuscript after reading their comments carefully.

Reviewer 1 ·

Basic reporting

no comment

Experimental design

no comment

Validity of the findings

no comment

Reviewer 2 ·

Basic reporting

no comment

Experimental design

no comment

Validity of the findings

I am still not convinced about the usefulness of the proposed method. Several deep learning based methods have developed for predicting RNA secondary structures. These methods are a lot more accurate than RNAplfold. However, the overall performance of PredPair, the method developed here is a lot worse than RNAplfold based on the precision-recall curve. Any conclusion drawn from such a low performing method is unlikely giving any insight.

·

Basic reporting

L101 "While this network based on the first principles naturally performs"
In general, the word "first principle" refers to physics based models (rules need not to be supported by any data).
Thus, it sounds strange to use it for methods such as neural networks, which is an extreme opposite to the first principle (deriving rules from data).
I recommend different words such as "data driven".

* L114 "regularization L2"
L2 regularization

* Fig 1 "The main layers of PredPair are indicated in yellow"
There is no part depicting layers nor part colored in yellow.
Please check.

* L130 "the fifth dimension to the vector containing '2' for the query position in the "question sample" or the correct pairing in the "answer sample" and '1' for all remaining positions."
In Fig 1, the query position is represented as '1' not '2', contradicting this explanation.
Please make them consistent.

* L260 "state-of-the-art method RNAplfold"
It sounds strange to refer RNAplfold as "state-of-the-art".
Thermodynamics-based methods such as RNA*fold are less accurate than recent machine learning based methods.
Moreover, RNAplfold is even not the best among thermodynamics-based methods.
I recommend different words such as "standard method" or "widely-used method".

* L288 "p-value << 0.01"
Please decribe the exact p-value.

* Fig 4
The range of x and y axes are trimmed arbitrarily.
Please keep the range as [0.0, 1.0], and if necessary, prepare an inset magnifying the part of interest.

* Fig 6
The authors compare a prediction of RNAfold (single structure) and a basepair matrix of PredPair (that can contain the information of multiple structures).
Thus, I feel the comparison is not fair-minded, making the authors' augument invalid.
Please use a basepairing probability matrix of RNAfold for better comparison.

* Fig 6 "with the scale given above"
The color scale bar is aside, not above.
Please correct.

* L317 "while incorrect do not"
while incorrect predictions do not

* Fig 11
This figure has many problems:
1) In the legend, the same families are represented by different symbols/colors.
e.g. two "GA-cis_RNA" in 9-th and 10-th lines.
There are many other examples of the same problem.
Please correct or explain why this can happen.
2) While the caption says 53 families are plotted, the number of points in Fig 11A is clearly larger than 53.
I wonder maybe multiple sequences are plotted from one family.
But if all sequences belonging the families are plotted, the number of points seems too small.
Please explain this.
3) The discussion of Fig 11B is insufficient.
According to the main text, this analysis was performed to investigate whether learned features represent structure information.
The results show that there are no colored cluster appeared in Fig 11B, suggesting that learned features DO NOT represent structure information.
However, the authors did not mention this critical issue, suddenly stopping their discussion (L353-355).
Please provide more discussion on what Fig 11B suggests.

* Typos and inconsistent hyphenation
"tSNE" or "t-SNE"
"base-pair" or "base pair"
Many others; please use the proofreading service.

Experimental design

no comment

Validity of the findings

According to Abstract, the main aim of this study is to interpret learned features.
However, the results most relevant to this aim, Fig 11, has many problems as described above.
Thus, I cannot believe the validity of the conclusion in the current form.

Reviewer 4 ·

Basic reporting

In this paper, the authors propose a method called PredPair that directly predicts base pairs in a sequence without using the free energy of substructures.
The authors analyzed the internal representation for secondary structure prediction by building a neural network without prior knowledge.
I enjoyed reading this paper. I hope the authors will address some points shown below to improve the quality of their study.

Experimental design

There is no description as to why this model architecture was adopted. It should mention why the combination of these functions in this order is useful for RNA secondary structure prediction. The current description only tells that it is a combination of well-known architectures.

The method of predicting secondary structure is unclear. In the paper, the certainty matrix is the output of the neural network, which is used to predict the secondary structure. However, if a threshold is set for the certainty matrix, there may be cases where a particular base forms base pairs with multiple bases.
If such a problem arises, how does the author deal with it?

There is only one comparison tool, RNAplfold. I understand the authors' argument that the accuracy of secondary structure prediction is not strongly emphasized in this paper. However, the number of comparison tools should still be increased. For example, comparisons should be made with deep learning based tools similar to this method. SPOT-RNA, which is presented within this paper, seems to be an appropriate comparison because it outputs something like base pair probabilities.

In Figure 10, the frequency of each pair is compared, but this appears to be an obvious result. It seems that the frequency of pairs in the training data is simply reflected directly in the test. I do not understand what the authors are trying to assert in this section.

Validity of the findings

The results are clearly presented and the data and programs used within the paper are publicly available.

---

## Round 0.3 · Major Revisions

Of the two reviewers who previously gave the comments for further revision, only one of them has agreed to review the revised manuscript. As you can see from the comments below, the reviewer points out that the main text has not been adequately revised to reflect the updated results (figures/abstract). Please read the comments carefully and adjust the manuscript accordingly. Thanks for your patience, in advance.

·

Basic reporting

* Fig 6
In the revised figure, the basepairing probability matrix of RNAfold indicates that it does predict the hairpins as suboptimal structures.
Thus, the authors should update their discussions in the main text e.g.
L314. "Here, RNAfold mispredicted the hairpins (Figure 6, B)."
Please check all other corresponding parts in the main text.

* Fig 11A
The revised figure still has problems e.g. there are two "Mu-gpT-DE_RNA" legends in different colors.
Please check again carefully.
In addition, I wonder why the distribution of plots is different from the previous version.
The authors just changed the labeling, thus the distribution of plots should be the same as the previous version.

* Abstract
Based on the revised Fig 11B, learned features do not represent secondary structures well.
Thus, the authors should tone down their conclusion e.g.
Abstract. "The embeddings of RNA sequences created by PredPair tend to reflect underlying biological properties"
Please check all corresponding parts in the main text.

Experimental design

no comment

Validity of the findings

Based on the revised Fig 6 and Fig 11, the authors need to modify their conclusion from the current form.

---

## Round 0.4 · accepted · Accept

Now that the remaining referee recommends the acceptance of your latest manuscript, I am happy to inform you that I will recommend its acceptance to the section editor. Congratulations!

·

Basic reporting

no comment

Experimental design

no comment

Validity of the findings

no comment